# Maximal Reduction of STIC Acquisition Time for Volumetric Assessment of the Fetal Heart—Benefits and Limitations of Semiautomatic Fetal Intelligent Navigation Echocardiography (FINE) Static Mode

**DOI:** 10.3390/jcm11144062

**Published:** 2022-07-14

**Authors:** Michael Gembicki, Jann Lennard Scharf, Christoph Dracopoulos, Amrei Welp, Jan Weichert

**Affiliations:** Department of Gynecology & Obstetrics, Division of Prenatal Medicine, University Hospital of Schleswig-Holstein, Campus Luebeck, 23538 Luebeck, Germany; jannlennard.scharf@uksh.de (J.L.S.); christoph.dracopoulos@uksh.de (C.D.); amrei.welp@uksh.de (A.W.); jan.weichert@uksh.de (J.W.)

**Keywords:** 4D, cardiac, fetal echocardiography, spatiotemporal image correlation, STIC, ultrasound, semiautomatic navigation

## Abstract

(1) Objective: To scrutinize the reliability and the clinical value of routinely used fetal intelligent navigation echocardiography (FINE) static mode (5DHeartStatic™) for accelerated semiautomatic volumetric assessment of the normal fetal heart. (2) Methods: In this study, a total of 296 second and third trimester fetuses were examined by targeted ultrasound. Spatiotemporal image correlation (STIC) volumes of the fetal heart were acquired for further volumetric assessment. In addition, all fetal hearts were scanned by a fast acquisition time volume (1 s). The volumes were analyzed using the FINE software. The data were investigated regarding the number of properly reconstructed planes and cardiac axis. (3) Results: A total of 257 volumes were included for final analysis. The mean gestational age (GA) was 23.9 weeks (14.3 to 37.7 weeks). In 96.9 (standard acquisition time, FINE standard mode) and 94.2% (fast acquisition time, FINE static mode) at least seven planes were reconstructed properly (*p* = 0.0961, not significant). Regarding the overall depiction rate, the standard mode was able to reconstruct 96.9% of the planes properly, whereas the static mode showed 95.2% of the planes (*p* = 0.0098). Moreover, there was no significant difference between the automatic measurement of the cardiac axis (37.95 + 9.14 vs. 38.00 + 8.92 degrees, *p* = 0.8827, not significant). (4) Conclusions: Based on our results, the FINE static mode technique is a reliable method. It provides similar information of the cardiac anatomy compared to conventional STIC volumes assessed by the FINE method. The FINE static mode has the potential to minimize the influence of motion artifacts during volume acquisition and might therefore be helpful concerning volumetric cardiac assessment in daily routine.

## 1. Introduction

Congenital heart disease (CHD) is the most common birth defect [1,2,3,4]. CHD is also the leading cause of neonatal morbidity and mortality derived from congenital birth defects [4,5]. It is known that up to 90% of CHD cases occur in a low-risk population, which makes the prenatal diagnosis of CHD a major challenge [4,6,7]. Therefore, as many pregnancies as possible should undergo targeted prenatal sonographic screening for CHD [8,9,10,11].

Still, the prenatal detection rates of CHD by conventional screening methods remain low, as shown by detection rates ranging from 15 to 39% [12,13]. Various factors make the prenatal diagnosis of CHD difficult, e.g., the complex anatomy of the heart, its small size and its motion [11]. Especially in untrained hands, the fetal heart may be the most difficult organ to examine by ultrasound [14].

New solution approaches are needed in order to solve these problems. Several new methods have therefore been proposed. In addition to expensive, time-consuming and relatively insufficient training programs, technical solutions might be useful. Specifically, four-dimensional (4D) ultrasound with spatiotemporal image correlation (STIC) has been shown to overcome some of the mentioned difficulties [4,15,16,17,18] and has the potential to increase the detection rate of CHD [16,19,20].

Nevertheless, the analysis of a STIC volume remains relatively difficult [11]. A STIC volume displays a cine loop of a complete single cardiac cycle in motion [21] and contains all the information needed to thoroughly examine the fetal heart [22]. Unfortunately, the usual manual depiction of this particular information is highly operator-dependent and lacks standardization [11,23]. A profound understanding of the fetal cardiac anatomy is mandatory, especially when the heart is abnormal [11,24,25,26].

In the recent past, a semiautomatic algorithm including artificial intelligence has been introduced to face the challenges of STIC volume analysis [11]. The identification and selection of key anatomical landmarks allows the fetal intelligent navigation echocardiography (FINE) technique to automatically display the specific diagnostic planes [23,27] needed for a detailed fetal echocardiogram [28,29]. This is achieved by using operator-independent algorithms that are predictable and adaptive [4,23]. With this technique the necessary diagnostic information can be extracted automatically and efficiently [30].

FINE has proven to be able to generate the nine fetal echocardiography views in up to 100% of STIC volumes derived from normal cases [11,24,25,30]. Manual manipulation of the STIC volumes and cardiac planes is not required [4]. Thus, FINE simplifies the fetal echocardiography and reduces operator dependency (FINE is commercially known as 5DHeart^TM^). In cases of CHD, FINE was able to show a sensitivity of 98% and a specificity of 93% [31]. Therefore, it has been suggested that FINE could be implemented as a fetal echocardiography screening tool [4,31]. Recent studies added color and power Doppler to FINE and showed useful results about cardiac structure and function in both normal and abnormal fetal hearts [32,33].

Currently, our group was able to show the effectiveness of this technique in unexperienced hands [30] as well as in unfavorable fetal spine localizations [34]. Moreover, other groups have demonstrated that FINE works well in generating three specific abnormal cardiac views in cases of fetal D-transposition of the great arteries [35]. This is comparable in accuracy to conventional 2D fetal cardiac examination in normal second trimester fetuses with a significant reduction in examination time [36].

The quality of FINE is based on the quality of the STIC volumes used for analysis (e.g., fetal spine position, minimal or absent shadowing, absent fetal breathing, hiccups or movement, and adequate image clarity) [11]. In particular, fetal movements can make it very difficult to obtain a high-quality STIC volume. Usually, a mechanical ultrasound probe needs 9–12 s to record a STIC volume. Fetal movements can occur during that time and force the investigator to reset the process and obtain another volume, which consumes time. With the aim of improving the FINE method further, new features have now been introduced, e.g., the static mode [4]. The three-dimensional static volume is obtained in a very rapid acquisition time (i.e., 1 s) and is characterized by a high frame rate. The hereby generated nine echocardiography views are static and without motion. The nine echocardiography views generated by FINE standard and static mode are shown in Figure 1. In the present study, we want to investigate the reliability and the clinical value of routinely used FINE static mode for accelerated semiautomatic volumetric assessment of the normal fetal heart.

## 2. Materials and Methods

### 2.1. Subjects

All women undergoing second and thirdtrimester ultrasound at the Women’s University Hospital of Schleswig-Holstein, Campus Luebeck, are routinely investigated by additional 3D and 4D echocardiographic examination with STIC volume acquisition compared to conventional 2D fetal echocardiography. Those volumes are stored and used for both onsite processing and future offsite analysis by FINE. The evaluation was approved by the local ethics committee. The volumes used for this study were acquired between September 2019 and January 2022. We examined a total of 296 fetuses during second and thirdtrimester targeted ultrasound. For this study, we excluded abnormal hearts. The included volumes were obtained by two expert investigators (J.W. and M.G.) and had to match certain quality requirements (e.g., minimal or absent shadowing, a clearly visible transverse aortic arch, absent fetal breathing, hiccups or movement, and adequate image clarity). The quality was judged by one expert investigator (J.W.).

### 2.2. Acquisition of STIC Volumes

All volumes included in this study were acquired by two physicians, being experts in fetal echocardiography, using a Samsung Hera W10 device (Samsung Medison, Seoul, Korea). The volumes were recorded starting from the four-chamber view using a mechanical convex transducer (CV1-8A, 1 to 8 MHz, Samsung Medison, Seoul, Korea) by performing automatic transverse sweeps through the fetal chest. The acquisition time for the conventional STIC volumes ranged from 9 to 12 s. In addition, all fetal hearts were scanned by a fast acquisition time volume (acquisition time 1 s). The acquisition angles for both kinds of volumes ranged from 15 to 35°, depending on gestational age.

### 2.3. Examination with FINE and FINE Static Mode

The analysis of the acquired volumes took place on the ultrasound machine using the installed FINE software by the same investigators. FINE semiautomatically generates nine standard fetal echocardiography planes ((1) four-chamber view; (2) five-chamber view; (3) left ventricular outflow tract; (4) short-axis view of great vessels/right ventricular outflow tract; (5) three-vessel–trachea view; (6) abdomen/stomach; (7) ductal arch; (8) aortic arch; and (9) superior and inferior vena cava). In order to generate the planes, the operator was instructed by the software to mark seven anatomical landmarks of the fetal heart ((1) cross-section of the aorta at the level of the stomach; (2) cross-section of the aorta at the level of the four-chamber view; (3) crux; (4) right atrial wall; (5) pulmonary valve; (6) cross-section of the superior vena cava; and (7) transverse aortic arch), resulting in a complete reconstruction of the nine diagnostic views of the fetal heart. This marking is guided by the software (e.g., written instructions on the screen, illustrating pictures of the landmarks, and automatic presentation of the correct planes).

The process of analyzing the fast-acquired volumes (3D, FINE static mode) works exactly the same way. The only difference is that the volumes contain no movement of the fetal heart, in contrast to the normal volumes, which are in motion. The process of generating the nine diagnostic planes using FINE and FINE static mode is illustrated in the Appendix A.

In addition, both modes automatically measure and display the angle of the fetal cardiac axis.

### 2.4. Virtual Intelligent Sonographer Assistance (VIS-Assistance)

VIS-Assistance allows user-independent sonographic exploration of the surrounding structures in each of the nine cardiac diagnostic views (virtual sonographer) [11]. The volumes are automatically scanned in a targeted manner in the form of a videoclip with the aim of obtaining specific structures [11]. VIS-Assistance improves the success rate of obtaining the fetal echocardiography view of interest and provides more information about the diagnostic plane and its surrounding structures [11].

### 2.5. Analysis by the Investigators

All volumes were analyzed using the FINE software and rated by the expert panel regarding the number of those properly reconstructed. These were subsequently compared to those derived from the static approach. For each case, the overall image quality was assessed from “very good”, “good” and “moderate” to “poor”.

### 2.6. Statistics

The data were investigated regarding the number of properly reconstructed planes and cardiac axis. GraphPad Prism 9 for Mac (Version 9.40, GraphPad Software Inc., La Jolla, CA, USA), GraphPad QuickCalcs (GraphPad Software Inc., La Jolla, CA, USA), and Microsoft Excel 2016 for Mac (Version 16.61.1, Microsoft Corp., Redmond, WA, USA) were used. Descriptive statistics, *t*-tests and McNemar tests were applied. A statistical level of *p* < 0.05 was assumed to be significant.

## 3. Results

In total, 296 fetuses were investigated by standard and fast acquisition time STIC. We excluded 39 cases, of which 22 had abnormal hearts, 7 were in the first trimester of pregnancy, and 10 had incomplete data. A total of 257 volumes were included for final analysis. The mean gestational age (GA) was 23.9 weeks (14.3 to 37.7 weeks) and the mean BMI at scanning date was 27.5 kg/m^2^ (18.6 to 65.3 kg/m^2^). The cases were rated regarding image quality, of which 10.1% (*n* = 26) were rated “very good”, 50.2% (*n* = 129) were rated “good”, 37.7% (*n* = 97) showed “moderate”, and 2.0% showed (*n* = 5) “poor” image quality.

In 96.9 (standard acquisition time, FINE standard mode) and 94.2% (fast acquisition time, FINE static mode) at least seven planes were reconstructed properly (*p* = 0.0961, not significant). Regarding the overall depiction rate, the standard mode was able to reconstruct 96.9% of the planes properly, whereas the fast mode showed 95.2% of the planes (*p* = 0.0098). The depiction rates for both modes are shown in Table 1.

The highest drop-out rates were found for the sagittal planes. The drop-out rates for the ductal arch were 8.2 and 8.6% (standard and static mode), 7.4 and 5.8% for the aortic arch, and 4.3 and 6.2% for the superior and inferior vena cava, respectively. In addition, the static mode showed a high drop-out rate for the five-chamber view (7.0%). The drop-out rates for all planes and both modes are shown in Table 2.

Moreover, no significant difference according to the automatic measurement of the cardiac axis could be detected between the two different modes (37.95 + 9.14 vs. 38.00 + 8.92 degrees, *p* = 0.8827, not significant). Mean and standard deviation were not significantly different, and the results of both modes passed the normality test. The distribution of the cardiac axis angles is shown in Figure 2, indicating a similar distribution pattern.

## 4. Discussion

In this study we were able to show some important results concerning FINE and FINE static mode. We were able to demonstrate the reliability of FINE within our current study sample. The correct depiction rate of at least seven planes using FINE was 96.9%. This is consistent with the results derived from other studies [11,24,25,30,34]. In our previous study, the overall depiction rate of the diagnostic planes was 83.3% to 95.2% using FINE alone and 96.7% to 100.00% with the additional use of VIS-Assistance [30]. According to the literature above, FINE was repeatedly able to show depiction rates up to 100%.

We could demonstrate the feasibility of FINE static mode for the first time. The correct depiction rate of at least seven fetal echocardiographic planes using the static mode was 94.2%. There was no significant difference to the standard mode in our population regarding the proper image reconstruction of at least seven diagnostic planes. The automatically calculated cardiac axis also did not differ. Although the overall depiction rates of 96.9 (standard mode) and 95.2% (static mode) showed a statistically significant difference, the difference is small enough to be rated as not clinically relevant.

FINE static mode might be able to facilitate volume acquisition and plane reconstruction facing very active fetuses, saving time and without losing accuracy. This could be useful regarding the intention to establish FINE as a screening tool in daily routine. FINE static mode might need a lower number of sweeps to acquire a sufficient volume in comparison to the standard mode. We include FINE and FINE static mode planes derived from moving fetuses in the Appendix A.

We think that the implementation of automatization and artificial intelligence in diagnostic processes such as fetal echocardiography is very promising. As mentioned above, the prenatal detection rates of CHD by conventional screening methods are low, as shown by detection rates from 15 to 39% [12,13]. The current guidelines and approaches of teaching 2D fetal echocardiography might not be sufficient to improve that rate. According to these findings, novel methods are needed. Yeo et al. showed a sensitivity of 98% and a specificity of 93% using FINE in cases of CHD [31]. Yeo et al. also demonstrated the ability of FINE to prenatally detect dextrocardia with complex CHD and two cases of tetralogy of Fallot (TOF) with pulmonary atresia [26,37]. In addition, in 34 cases of D-TGA (d-transposition of the great arteries), the FINE method showed a high success rate in generating specific abnormal cardiac views and has therefore been promoted to work as a screening tool for this congenital defect. In that work, 85.7% of the STIC volume data sets showed two or three of the abnormal cardiac views (left ventricular outflow tract, right ventricular outflow tract, and three-vessel and trachea view) [35]. Another study was able to point out the display rates of the same specific abnormal fetal echocardiography views in 25 cases of double-outlet right ventricle (DORV), showing comparable results and conclusions [38].

Furthermore, FINE demonstrated its abilities to work well with different features and in different situations. For example, Yeo et al. added color and power Doppler to FINE, which showed promising results [33]. The rate of successfully generating eight fetal echocardiography views with accurate color and S-flow Doppler information was 89–100% [33]. The images obtained using color Doppler FINE allowed a precise diagnosis of fetal hypoplastic left heart and coarctation of the aorta at 26 weeks of gestation [32].

In one of our previous works, we investigated the use of FINE in different states of experience in fetal echocardiography and showed that experts, advanced practitioners and beginners in fetal echocardiography were able to adequately perform FINE in a short period of time (21–74 s per investigation), starting from an existing STIC volume [30]. In addition, we were able to show the accuracy of FINE in fetuses with unfavorable spine positions in another work [34]. In an optimal fetal spine position between 5 and 7 o’clock, 94.9% of the diagnostic planes were displayed properly [34]. The correct depiction rates in the other groups ranged from 92.4% to 87.3% [34]. According to the idea that FINE could be used as a general screening tool for CHD, a depiction rate of at least 87.3%, even in unfavorable fetal positions, is promising.

Among other authors, the value of 4D fetal echocardiography is often doubted. Some of them criticize a high user dependency and the absence of standardization. Novaes et al. showed that STIC volume acquisition was successful in 97.3% of patients, but all planes required for optimal fetal heart screening were seen in only 49% of their volumes [39]. The conclusion was that STIC might be used as a tool to improve the screening for CHD, but practice remains the most important factor for the image quality of STIC volumes and consecutive planes [39]. There are many reported success rates in obtaining STIC volumes in normal fetuses. The values range widely, from 26 to 100%, and the success rate depends on several conditions, e.g., fetal spine position, maternal BMI, and gestational age [14,15,17,24,40,41,42].

Roberts proposed two possible ways that4D echocardiography can be used to improve the detection rates of CHD on a large scale: (i) acquisition of STIC volumes locally at the screening center with subsequent analysis by a remote expert in fetal echocardiography; and (ii) storage and analysis of the volumes by the same examiner at a later time [43]. Roberts highlights that a general screening program with STIC technology would have to involve a hybrid of the models described above, based on a priori risk [43].

In our opinion, the ability to obtain a sufficient STIC volume and analyze it with FINE might be learned easier than performing accurate 2D fetal echocardiography. As we have shown, FINE works well in unexperienced hands [30]. By comparing the generated planes to the normal anatomy, the investigator might be able to detect altered anatomy based on their general training. The addition of electronic guidance (e.g., comparison pictures on the screen or automatic alerts) will help to do so. It is very important to point out that in general screening it is not necessary to define the exact cardiac malformation, but only to detect it.

The potential use of 4D echocardiography is not accepted by everybody. We think that new technical solutions might be very helpful, because the detection rates of CHD vary broadly and remain low in a general setting. According to the ISUOG guidelines, some of these variations can be conducted to different levels of examiner expertise [29]. Intense training for investigators is important to improve the effectiveness of a screening program. Hunter et al. showed that the detection rate of major cardiac anomalies doubled after the implementation of a 2-year training program at a medical facility in Northern England [44]. In our eyes, techniques that are more operator-independent might be more effective in terms of improving detection rates as well as saving time and money.

Our study has strengths and limitations. On the one hand, the relatively large sample size is a strength, especially because the volumes have been obtained under real-life conditions and might therefore be representative of other facilities. Our work is the first to show the practicability of FINE static mode in a routine setting. The volumes and corresponding generated planes of FINE static mode are not moving. As we have seen, there is no clinically relevant difference in the depiction rate compared to conventional STIC volumes processed by FINE. Nevertheless, it is conceivable that the detection or correct diagnosis of CHD might be limited because moving structures provide more information. In addition, FINE static mode cannot be combined with color Doppler. FINE static mode must prove its usefulness in detecting CHD in future studies, because we did not include cases of CHD. Another limitation is the expert acquisition of the volumes. In order to be used as a screening tool, the acquisition of STIC volumes must be reliable in non-expert hands as well. As we have demonstrated, the FINE postprocessing itself works in the hands of a beginner [30]. In addition, we had a certain selection bias, because not all patients presenting within the study period were scanned with 3D/4D ultrasound. The retrospective character of our study can be seen as a limitation, although we were able to demonstrate the real-life conditions in that way.

To conclude, based on our results, FINE static mode is a reliable method. It provides similar depiction rates of the cardiac anatomy compared to conventional STIC volumes assessed by conventional FINE. FINE static mode has the potential to minimize the influence of motion artifacts during volume acquisition and might therefore be helpful when concerning volumetric cardiac assessment in daily routine. Evaluation of the fetal heart by FINE might be useful in facilitating the detection of fetal cardiac anomalies during general screening and, therefore, raise the detection rates of CHD. Future studies should aim to demonstrate the feasibility and validity of the complete workflow from volume acquisition to postprocessing via FINE and FINE static mode in less experienced hands, first-trimester fetuses and fetuses with CHD to prove the use of FINE as a screening tool.

## Figures and Tables

**Figure 1 jcm-11-04062-f001:**
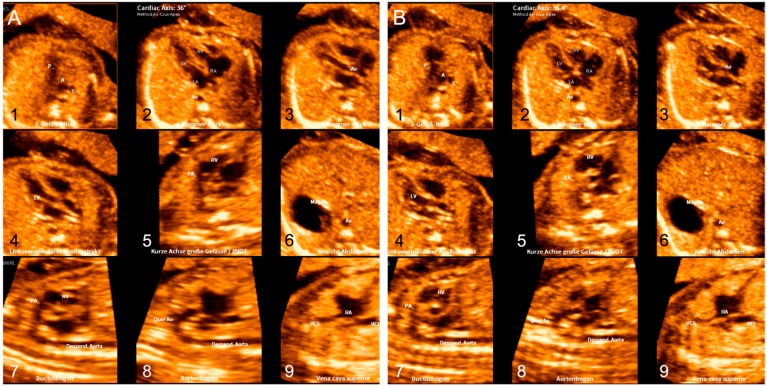
The nine diagnostic planes of the fetal heart reconstructed by FINE standard mode (**A**) and FINE static mode (**B**). (1) Three-vessel–trachea view; (2) four-chamber view; (3) five-chamber view; (4) left ventricular outflow tract; (5) short-axis view of great vessels/right ventricular outflow tract; (6) abdomen/stomach; (7) ductal arch; (8) aortic arch; and (9) superior and inferior vena cava.

**Figure 2 jcm-11-04062-f002:**
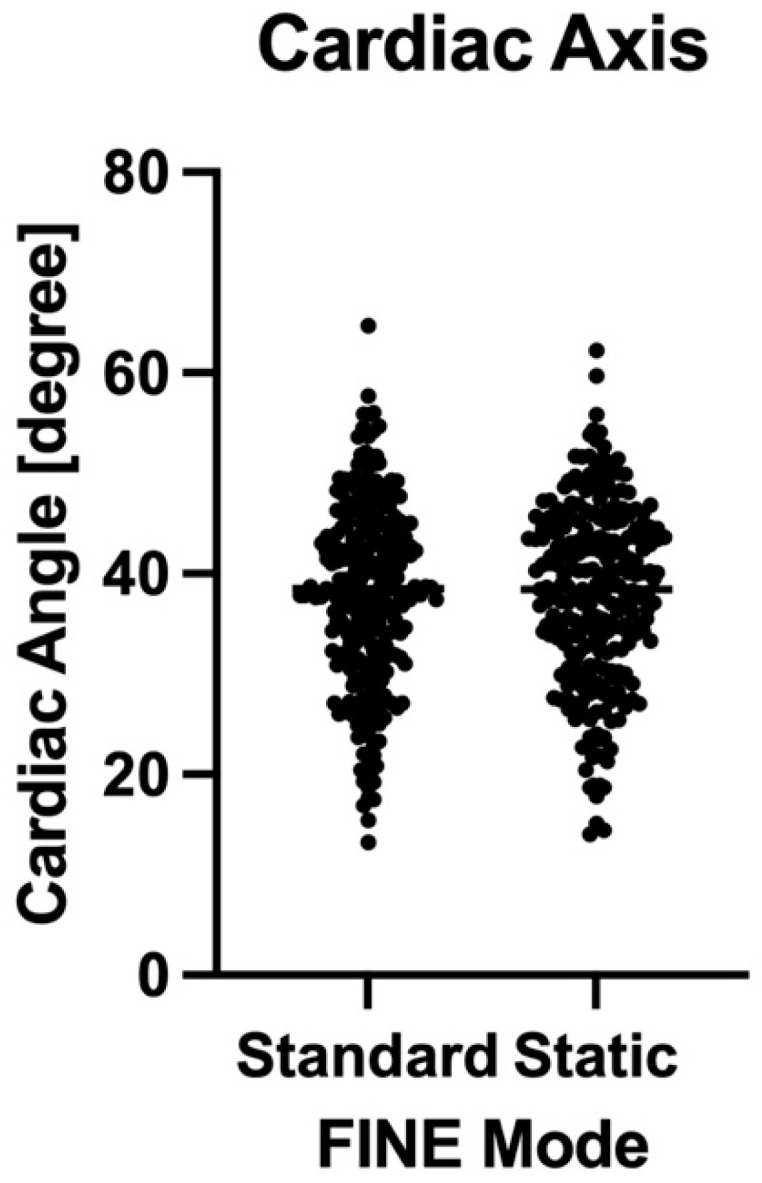
Distribution of the cardiac axis angles measured by FINE standard and static mode.

**Table 1 jcm-11-04062-t001:** Properly displayed planes for both modes.

**Mode**	**Depiction Rate ≥7 Planes (% of All Cases)**	***p*-Value**
Standard	96.9	0.0961
Static	94.2	
**Mode**	**Overall Depiction Rate (% of All Planes)**	***p*-Value**
Standard	96.9	0.0098
Static	95.2	

**Table 2 jcm-11-04062-t002:** Drop-out rates for each plane.

Plane	Standard Mode Drop-Out Rate (%)	Static Mode Drop-Out Rate (%)
Three-vessel–trachea view	3.1	4.7
four-chamber view	0.0	0.8
five-chamber view	1.2	7.0
left ventricular outflow tract	2.3	5.1
right ventricular outflow tract	1.9	4.3
abdomen	0.0	1.2
ductal arch	8.2	8.6
aortic arch	7.4	5.8
vena cava	4.3	6.2

## Data Availability

The data presented in this study are available on request from the corresponding author. The data are not publicly available due to data privacy restrictions.

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
