# Peer review of "Maximal Reduction of STIC Acquisition Time for Volumetric Assessment of the Fetal Heart—Benefits and Limitations of Semiautomatic Fetal Intelligent Navigation Echocardiography (FINE) Static Mode"

_jcm, 2022, doi:10.3390/jcm11144062_

Round 1

Reviewer 1 Report

In this retrospective study the authors evaluate the reliability and the clinical value of routinely used fetal intelligent navigation echocardiography (FINE) static mode (5DHeartStaticTM) for accelerated semiautomatic volumetric assessment of the normal fetal heart. The authors conducted a well-designed study with a adequate number of cases and a clearly selected study group. The manuscript is written in a straight-forward manner. In addition, there is a well-developed image and video material, which describes the technology very well.

Minor corrections

Page 6, Line 193: “The distribution of the cardiac axis angles is shown in Figure 2.“ Please describe the results on this issue more precisely in the result section.

Page 6, Line 211: “The automatically calculated cardiac axis did not differ as well (37.95 + 9.14 vs. 211 38.00 + 8.92 degrees, p = 0.8827, not significant).“ Authors should describe this result in the result section!

Page 7, Line 239: “Another paper […]”. Please rephrase “Another study”

The discussion part seems a bit long. Perhaps this can still be shortened at appropriate points.

Author Response

Dear Sir or Madam,

Thank you very much for your input. We answered your comments as following:

Page 6, Line 193: “The distribution of the cardiac axis angles is shown in Figure 2.“ Please describe the results on this issue more precisely in the result section.

Answer: We described those results more precisely.

Page 6, Line 211: “The automatically calculated cardiac axis did not differ as well (37.95 + 9.14 vs. 211 38.00 + 8.92 degrees, p = 0.8827, not significant).“ Authors should describe this result in the result section!

Answer: We did as suggested by the reviewer.

Page 7, Line 239: “Another paper […]”. Please rephrase “Another study”

Answer: We changed our wording as suggested.

The discussion part seems a bit long. Perhaps this can still be shortened at appropriate points.

Answer: Thank you very much for your comment. We shortened the discussion part.

Reviewer 2 Report

This is a very interesting study, designed to investigate the clinical value of FINE static mode as an accelerated semiautomatic evaluation of the normal fetal heart in the second and third trimesters.

The authors demonstrate FINE static mode as a reliable method that provides similar depiction rates of the cardiac anatomy planes compared to conventional STIC / FINE volumes, but with the advantages of a time-saving technique, less dependency on expertise, and minimization of the motion artifacts' adverse effects.

Based on the study results and previous findings, the authors nicely present the feasibility of FINE static mode for daily routine use to screen for fetal heart abnormalities and its potential to increase the CHD detection rates.

Author Response

Dear Sir or Madam,

Thank you very much for your input. We're always giving our best to publish high quality data and are very proud of our work. Your encouraging review is very welcome!

Kind regards!